# Do Patients Residing in Provincial Areas Transport and Spend More on Cancer Treatment in Korea?

**DOI:** 10.3390/ijerph18179247

**Published:** 2021-09-01

**Authors:** Woorim Kim, Kyu-Tae Han, Seungju Kim

**Affiliations:** 1National Cancer Center, Division of Cancer Control & Policy, National Cancer Control Institute, Goyang 10408, Korea; wklaura@gmail.com (W.K.); kthan.phd@gmail.com (K.-T.H.); 2Department of Nursing, College of Nursing, The Catholic University of Korea, Seoul 06591, Korea

**Keywords:** medical travel, healthcare costs, healthcare utilization, regional disparity, treatment patterns

## Abstract

Background: With the increasing burden of cancer worldwide, a need exists to investigate patterns of healthcare utilization and costs. This study aimed to investigate whether the area of residence is associated with the likelihood of a patient receiving treatment at an institution located outside their residing region. This study also analyzed whether medical travel was related to levels of healthcare utilization and costs. Methods: This study used the 2007 to 2015 National Health Insurance (NHI) claims data. The residing area was categorized into capital area, metropolitan cities, and provincial area. Healthcare utilization was measured based on days of care and costs based on direct, covered medical costs. Chi-square test and analysis of variance (ANOVA) was conducted to investigate the general characteristics of the study population. The relationship between the dependent and independent variables were analyzed using the generalized estimating equation (GEE) model. Results: Of the 64,505 participants included in this study, 19,975 (31.0%) visited medical institutions located outside their residing area. Compared to individuals residing in the capital area, those living in provincial regions (OR 2.202, 95% CI 2.068–2.344) were more likely to visit medical institutions outside their residing area. Healthcare costs were higher in individuals receiving treatment at hospitals located elsewhere (RR 1.054, 95% CI 1.017–1.093). Conclusion: Cancer patients residing in provincial areas were likely to visit institutions located outside their residing area for treatment. Medical travel was associated with higher levels of spent healthcare costs. Policies should focus on preventing possible related regional cancer disparity and promoting optimal configuration of cancer services.

## 1. Introduction

Cancer is a leading cause of death worldwide and in many countries, the second cause of mortality in people aged below 70 years of age [1,2]. Importantly, whilst around 14.1 million new cancer cases are estimated to have occurred in 2012, cancer incidence is projected to further double by 2035 [3,4]. Unlike many non-communicable diseases which have shown trends of improvement, the burden of cancer is predicted to increase [5]. Cancer also incurs the highest number of deaths in Korea and the number of new cancer cases is rising [6]. The age standardized incidence and mortality rate were 264.4 and 76.6 per 100,000 individuals in 2017, with the prevalence of cancer escalating noticeably due to the recent improvements in survival rates [6]. Unsurprisingly, healthcare expenditures related to cancer constitutes a significant proportion of the National Health Insurance (NHI) budget of Korea, accounting for around 9% of total expenditures [7]. The total economic burden of cancer was estimated to exceed approximately $20,000 million USD (United States Dollar) in 2010 [8].

With the increasing burden of cancer in Korea, investigating patterns of healthcare utilization and medical costs has become increasingly important due to several reasons. First, Korea has a mandatory NHI system which covers around 98% of the entire population (excluding Medical Aid) that operates largely on a fee-for-based reimbursement system [9]. The referral system is weak, and patients are mostly able to visit a medical institution of their choice; although patients need a referral to visit specialized general hospitals, the level of barrier is not high [10]. Hence, cancer patients show a preference for large sized hospitals, in particular tertiary hospitals located in Seoul [11]. The preference is a partial result of tertiary hospitals being high volume centers equipped with different professional staff and higher quality [12]. However, movement of patients to specific areas for treatment requires monitoring because it may partially reflect disparities in treatment access, delays, and cost. For instance, patients visiting medical institutions located outside their residing area may face more difficulties in accessing treatment at an appropriate time, in addition to bearing additional costs from travel [11,13,14,15]. Therefore, a need exists to investigate whether patients’ area of residence affects the likelihood to receive treatment at a healthcare institution outside of their residing area and to further examine whether this difference affects levels of healthcare utilization and costs.

The aim of this study was to investigate whether gastric, colorectal, hepatocellular, and lung cancer patients’ area of residence (capital area, metropolitan cities, or provincial area) was associated with the likelihood of a patient to receive treatment at an institution located outside their residing region. Additionally, the aim of this study was to analyze whether medical travel was related to levels of healthcare utilization and costs. Healthcare utilization was measured based on the total days of care and costs based on total healthcare costs. The hypothesis was that patients residing in provincial areas would more likely receive treatment outside their residing area and that medical travel would be associated with higher levels of healthcare utilization and costs.

## 2. Methods

### 2.1. Data and Study Population

Data used in this study were the Korean national elderly sampled cohort database, which were collected using Korean National Health Insurance (NHI) claims data. This data consists of around 10% of individuals aged over 60 years at December, 2002 (N = 5.5 million people), whom were followed up until 2015 (sample size: approximately 550,000 during 2002–2015). Information on individual characteristics including demographic information, socio-economic information, healthcare utilization and treatment details, medical check-up, and medical institution are included.

This study aimed to identify the patient’s choice for major treatment sources by their residing area, and also investigate the association with healthcare utilization such as length of care and medical cost by difference between residential area and major treatment area in older aged cancer patients. In this study, we only included common cancer in South Korea such as gastric (C16), colorectal (C18–C20), lung (C33–C34), and liver (C22) cancer based on International Classification of Diseases, 10th revision (ICD-10) code for major symptom. Patients diagnosed before 2007 or patients with a history of being diagnosed with other cancer in the past five years were excluded. The data was aggregated by unit of every year from 1 to 5 years after first diagnosis to observe the changes over time (25,297 patients; gastric: 7742, colorectal: 7308, lung: 7152, and liver: 3095). Finally, the data used in this study consisted of 64,505 patient-years.

### 2.2. Outcome Measure

This study was conducted in two stages and hence had more than one dependent variable. The first dependent variable of this study was the location of a medical institution (local hospital vs. out-of-region hospital). This was because in the first stage, this study investigated the differences in selecting a medical institution according to where a patient resides. The residential area of a patient and the location of medical institution visited was defined based on the 18 administrative districts of Korea. If patients spent most of their medical costs at an institution located outside their residential area, these patients were categorized in the ‘out-of-region hospital’ group and vice versa.

In the second phase, the effect of the location of medical institution (local hospital vs. out-of-region hospital), a result of patient choice, on healthcare utilization was evaluated (Appendix A). Healthcare utilization was defined based on the sum of annual days of care and medical costs, namely the dependent variables of this study. As the elderly cohort data used in this study included individuals aged 60 years or above, we re-calculated the medical utilization based on the follow-up period by applying the formula below. The Korean Won (KRW)—USD (United States Dollar, $) exchange rate was applied to the calculation of medical costs ($1 = around KRW 1131.5 in 2015).
(1)Medical utilization (per year)=observed values (cost or days of care)÷observed period (days)×365 (days)

### 2.3. Interesting Variable

The interesting variables were separately considered by study phases, similarly with outcome variables. The first phase is based on the hypothesis that provincial patients tend to receive a treatment at hospital outside of residing area, and the primary variable of interest was the residing area of the patient. It was grouped into capital area (Seoul and Gyeonggi), metropolitan (Incheon, Daejeon, Daegu, Gwangju, Ulsan, and Busan), and others based on the administrative district. In the second phase, we investigated the healthcare utilization by difference between area of residence and major treatment area, thus, the difference of patient’s residential area and major treatment area was considered as major interesting variable, which was outcome variable in first phase. 

### 2.4. Covariates

Other independent variables were types of main treatment institution, sex (male, female), age (~69, 70 to 74, 75 to 79, or 80+ years), types of insurance coverage, economic status, period after first diagnosis (~1, 1 to 2, 2 to 3, 3 to 4, or 4 to 5 years), types of cancer, the Charlson comorbidity index (CCI), deaths in each observed year, and yearly trend. First, the types of main treatment institutions were defined based on the proportion of medical costs that were consumed by patients regarding which types of medical institution had the largest portion among total cost (general hospital, hospital, long-term care hospital, and others). Regarding classification of insurance coverage, around 97% of individuals are NHI beneficiaries in Korea, classified into the NHI self-employed and NHI employee groups. The NHI employee group includes all employees and employers, with their household members also being covered. NHI self-employed group includes all other individuals, with insurance premiums being calculated based on income, property, and living standards. The Medical-Aid group includes around 3% of low-income or disabled individuals who do not pay an insurance premium. Premiums are paid according to an individual’s economic status, classified based on deciles. After considering the distribution pattern of the study participants, this study classified this variable into the following groups: ~30 (low), 31 to 60 (mid-low), 61 to 80 (mid), 81 to 90 (mid-high), and 91+ percentiles (high). The CCI was utilized to incorporate clinical severity, calculated based on medical and symptom records during each year and excluded the score due to cancer. Deaths in each observed year were defined based on whether each patient died in the observed year. 

### 2.5. Research Ethical Approval

This study involves human participants but was not approved by an Institutional Board. Our research used secondary data, which is public data, and personal information, which is encrypted and cannot be distinguished.

### 2.6. Statistical Analysis

We first compared the regional distribution of the difference of the patients’ residential area and major treatment area by residing area based on mapping. Second, the distribution and general characteristics of the patients were measured using chi-square tests and analysis of variance (ANOVA) by the difference of patient’s residential area and major treatment area. Finally, multiple logistic regression analysis using the generalized estimated equation (GEE) model were conducted after controlling for all independent variables to investigate the association with treatment at the hospital outside of the residing area. Next, we also performed multiple gamma regression analysis using GEE model to investigate the impact of difference between the residing area and major treatment on medical utilization such as days of care and medical cost. All statistical analyses were performed using the SAS statistical software version 9.4 (Cary, NC, USA).

## 3. Results

The general characteristics of the study population are shown in Table 1. Of a total of 64,505 participants treated for gastric, colorectal, hepatocellular, and lung cancer, 19,975 (31.0%) visited medical institutions situated at a location different from their residing areas, whereas 44,530 (69.0%) participants visited medical institutions within their residing area. A total of 23,633 patients resided in the capital area, 14,064 patients in metropolitan cities, and 26,808 patients in provincial areas. The proportion of patients visiting another area for treatment was highest in those residing in provincial areas (38.8%), followed by those in the capital area (26.3%) and metropolitan cities (23.9%). Figure 1 provides a closer look at movement patterns among cancer patients. Overall, the proportion of patients receiving treatment at areas outside their residing region was higher in provincial regions. 

The average number of days of care and healthcare costs are shown in the Appendix A. In patients who received treatment in hospitals located inside their residing area, days of care were lowest in those residing in the capital area. In patients who received treatment in hospitals located outside their residing area, days of care were lowest in those residing in metropolitan cities (Figure 2).

The association between the participants’ area of residence and the location of hospital visited for treatment are presented in Table 2. Compared to individuals residing in the capital area, those in provincial regions (OR 2.202, 95% CI 2.068–2.344) were more likely to visit medical institutions elsewhere for treatment. Receiving treatment at a general hospital (OR 4.396, 95% CI 4.056–4.765) was also significantly associated with visiting an institution outside participants’ residing area.

Individuals receiving treatment at hospitals outside their residing areas (RR 1.054, 95% CI 1.017–1.093) tended to have higher levels of healthcare costs than participants treated in hospitals located inside their residing areas, as shown in Table 3. Similar tendencies were found regarding days of care, but the results were not statistically significant. 

## 4. Discussions

The results of this study show that gastric, colorectal, hepatocellular, and lung cancer patients living in provincial areas are more likely to visit a hospital located outside their residing area for treatment than individuals living in the capital area. However, such inclinations were not found in patients located at metropolitan cities. Receiving treatment at a general hospital was also significantly associated with the likelihood of medical travel. Higher amounts of healthcare costs were spent by cancer patients receiving treatment in areas outside their residing regions.

Many studies have investigated the relationship between patient outcomes and medical travel patterns. A systematic review which investigated the effect of travel distance (distance from the patient’s residing area to the treatment facility) on patient outcomes has shown mixed results, suggesting that healthcare facilities and a patient’s treatment options should be considered concurrently [16]. Patients living in rural areas eligible for Medicare tended to visit generalists in their local area, whereas they tended to travel to urban areas for specialist care [17]. Another study showed that healthcare utilization was lower in patients who needed to travel longer distances for healthcare [14], whereas others concluded that healthcare utilization and costs were higher in patients who lived further from a primary care physician [18]. Physician experience is a factor that can positively affect patient outcomes [19]. For instance, patients with ovarian cancer were found to have higher rates of mortality if they resided further away from a high-volume hospital [20]. These findings suggest that the medical travel patterns should not be defined simply based on absolute distance, but considered accounting for various factors such as patient severity and the distribution of resources.

An individual’s perception of a ‘better hospital’ can affect patient choice because cancer is a severe disease. In Korea, many general hospitals are skewed to the capital and metropolitan areas. Hence, as a result of patient selection, cancer patients residing in rural areas may favor care in institutions located outside their residing areas. The availability of high-end surgical technologies and hospital or physician reputation are also drivers of patient mobility [21]. In terms of hospital volume, individuals living in areas without large, prestigious hospitals nearby may prefer distant hospitals [22]. In fact, the congestion of cancer patients to the capital is a well-reported phenomenon in Korea. A study on gastric cancer patients revealed that despite gastric cancer occurring evenly in all geographical areas, most patients receive treatment in large volume institutions located in the capital area [23]. A large proportion of prostate cancer patients were also analyzed to travel for medical treatment irrespective of distance in a previous study [24]. The findings of this study add evidence on the topic of medical travel by exposing that patients residing in provincial areas have a particular tendency to travel outside their residing area.

Regional disparities in health have required attention in many countries. One factor that may act to increase the health gap between regions is the uneven distribution of resources. Imbalances in resource allocation can affect access to care, which may result in increased patient movement to hospitals outside their residing area for care [17]. The results of this study reveal that an equitable distribution of medical resources is important and provides several related implications. First, a high level of patient mobility and choice can inevitably lead to a certain amount of centralization, especially for specialized cancer services. This can be influenced by various national policies on health, and a favorable volume-outcome relationship have been demonstrated in the case of cancer [20,25]. However, the higher likelihood of patients residing in provincial areas to receive treatment at hospitals located outside their residing region requires monitoring and addressment because the findings show that healthcare costs tend to be higher in patients who travel for medical care. Such travelling may incur expenses arising from repeated diagnosis and tests, along with other costs that can arise due to various travel expenses and lost opportunity time [11]. Considering that the analysis of this study only accounted for direct healthcare costs, it can be assumed that actual total costs arising from medical travel would be significantly higher than the results shown. 

Second, the time required to travel longer distances that can offset the potential increase in patient outcomes requires further investigation. Receiving treatment at a distant hospital has been associated with reduced accessibility and increased time-to-treatment, which can escalate the risk of cancer progression [13,15,26]. Such tendencies may act as a source of regional cancer disparity, particularly in individuals with financial or physical constraints, such as the elderly and individuals with comorbidity, groups which also tend to have a higher risk of perioperative mortality [27,28]. The findings together infer the need to promote optimal configuration of cancer services that can account for patient healthcare utilization patterns, patient mobility, hospital capacity, and service quality [22]. To this end, policy makers will continuously monitor the healthcare utilization patterns of cancer patients, in addition to investigating policies that incline an efficient allocation of healthcare resources.

This study has some limitations. First, only direct, covered medical costs were accounted for in the analysis as information on non-covered medical costs and other expenses were unavailable in the data used. Second, this study could not account for the specific cancer stage due to data limitation. However, only first diagnosed patients were included in the study population to partially overcome this limitation. Third, several health-related behaviors, such as alcohol consumption and smoking, could not be considered as covariates. However, despite the limitations described above, this study offers important insights by revealing that cancer patients residing in provincial areas are most likely to medical travel for treatment and that healthcare costs tend to be higher in such patients using a large, nationally representative data.

## 5. Conclusions

In this study, we found that patients with gastric, colorectal, hepatocellular, and lung cancer living in provincial areas most often visited medial institutions located outside their residing region for cancer treatment. Medical costs were also higher in patients receiving care at hospitals located outside their residing areas. The findings infer the importance of appropriately distributing healthcare resources as individuals living in provincial areas may experience higher barriers in accessing cancer treatment. Continued efforts should be made to reduce the regional disparities in cancer.

## Figures and Tables

**Figure 1 ijerph-18-09247-f001:**
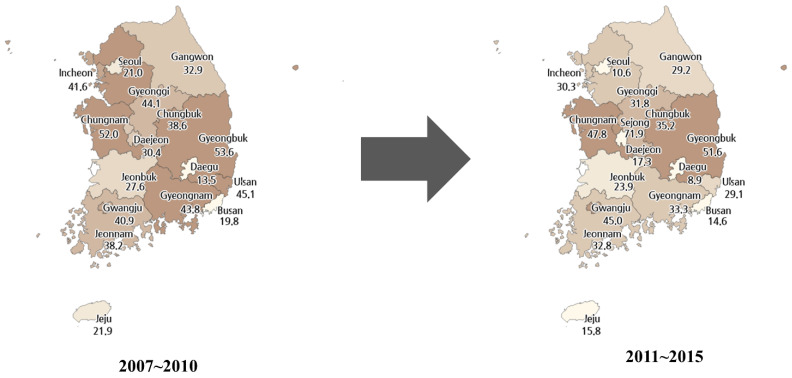
The difference of patient’s residing area and major treatment area.

**Figure 2 ijerph-18-09247-f002:**
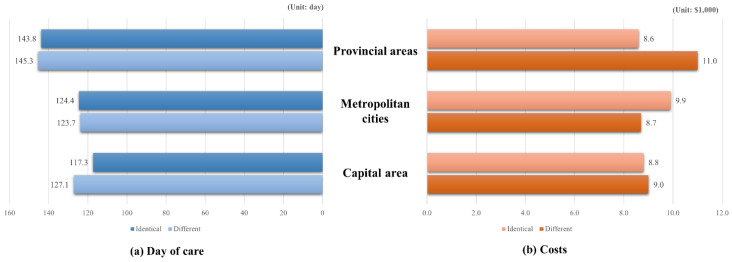
The average number of days of care and costs by residing area.

**Table 1 ijerph-18-09247-t001:** General characteristics of the study population.

Variables	Total	Area of Residence vs. Location of Hospital Visited for Treatment				
		Different		Identical		*p*-Value
		N	%	N	%	
Residing area						
Capital area	23,633	6208	26.3	17,425	73.7	<0.0001
Metropolitan cities	14,064	3366	23.9	10,698	76.1	
Provincial areas	26,808	10,401	38.8	16,407	61.2	
Institution of main treatment						
General hospital	44,108	16,507	37.4	27,601	62.6	<0.0001
Hospital	4641	1067	23.0	3574	77.0	
Long-term care hospital	2959	761	25.7	2198	74.3	
Clinic	12,797	1640	12.8	11,157	87.2	
Sex						
Male	36,991	11,781	31.8	25,210	68.2	<0.0001
Female	27,514	8194	29.8	19,320	70.2	
Age						
~69	7084	2694	38.0	4390	62.0	<0.0001
70~74	22,666	7643	33.7	15,023	66.3	
75~79	19,589	5727	29.2	13,862	70.8	
80~	15,166	3911	25.8	11,255	74.2	
Types of insurance coverage						
NHI self-employed	6333	1275	20.1	5058	79.9	<0.0001
NHI employed	18,084	5441	30.1	12,643	69.9	
Medical Aid	40,088	13,259	33.1	26,829	66.9	
Economic status						
Low	17,316	4444	25.7	12,872	74.3	<0.0001
Mid-low	11,525	3531	30.6	7994	69.4	
Mid	10,920	3499	32.0	7421	68.0	
Mid-high	10,619	3692	34.8	6927	65.2	
High	14,125	4809	34.0	9316	66.0	
Time since first diagnosis						
~1 y	25,297	8976	35.5	16,321	64.5	<0.0001
1~2 y	13,583	4264	31.4	9319	68.6	
2~3 y	10,545	3024	28.7	7521	71.3	
3~4 y	8422	2133	25.3	6289	74.7	
4~5 y	6658	1578	23.7	5080	76.3	
Diagnosed cancer type						
Gastric cancer	22,417	7068	31.5	15,349	68.5	<0.0001
Colorectal cancer	22,636	6739	29.8	15,897	70.2	
Hepatocellular carcinoma	6030	1951	32.4	4079	67.6	
Lung cancer	13,422	4217	31.4	9205	68.6	
Charlson Comorbidity Index (excluded Cancer)						
~2	33,904	10,301	30.4	23,603	69.6	<0.0001
3~5	15,337	4480	29.2	10,857	70.8	
6~	15,264	5194	34.0	10,070	66.0	
Died during observed year						
Yes	6564	1936	29.5	4628	70.5	0.0065
No	57,941	18,039	31.1	39,902	68.9	□
**Total**	64,505	19,975	31.0	44,530	69.0	□

NHI: National Health Insurance.

**Table 2 ijerph-18-09247-t002:** Association between area of residence and location of hospital visited for treatment.

Variables	Treatment at Hospital Outside of Residing Area			
	OR	95% CI		*p*-Value
Residing area				
Capital area	1.000	-	-	-
Metropolitan cities	0.938	0.866	1.015	0.1125
Provincial areas	2.202	2.068	2.344	<0.0001
Institution of main treatment				
General hospital	4.396	4.056	4.765	<0.0001
Hospital	2.118	1.891	2.373	<0.0001
Long-term care hospital	2.940	2.577	3.354	<0.0001
Clinic	1.000	-	-	-
Sex				
Male	1.000	-	-	-
Female	1.071	1.014	1.132	0.0141
Age				
~69	1.287	1.172	1.413	<0.0001
70~74	1.304	1.219	1.395	<0.0001
75~79	1.140	1.068	1.216	<0.0001
80~	1.000	-	-	-
Type of insurance coverage				
NHI self-employed	0.578	0.515	0.649	<0.0001
NHI employed	0.880	0.829	0.935	<0.0001
Medical Aid	1.000	-	-	-
Economic status				
Low	1.000	-	-	-
Mid-low	1.058	0.978	1.144	0.1589
Mid	1.097	1.010	1.192	0.0273
Mid-high	1.289	1.185	1.402	<0.0001
High	1.351	1.246	1.465	<0.0001
Time since first diagnosis				
~1 y	1.221	1.138	1.310	<0.0001
1~2 y	1.176	1.101	1.256	<0.0001
2~3 y	1.133	1.065	1.205	<0.0001
3~4 y	1.012	0.957	1.069	0.6841
4~5 y	1.000	-	-	-
Diagnosed cancer type				
Gastric cancer	1.000	-	-	-
Colorectal cancer	0.985	0.922	1.052	0.6510
Hepatocellular carcinoma	0.995	0.907	1.093	0.9223
Lung cancer	0.996	0.927	1.069	0.9035
Charlson Comorbidity Index (excluded Cancer)				
~2	1.114	1.055	1.176	0.0001
3~5	0.988	0.930	1.049	0.6844
6~	1.000	-	-	-
Died during observed year				
Yes	1.000	-	-	-
No	1.379	1.291	1.472	<0.0001
Year	0.930	0.919	0.940	<0.0001

**Table 3 ijerph-18-09247-t003:** Association between area of residence and days of care and healthcare costs.

Variables	Days of Care				Costs			
	RR	95% CI		*p*-Value	RR	95% CI		*p*-Value
Treatment at hospital outside of residing area								
Yes	1.021	0.986	1.058	0.2431	1.054	1.017	1.093	0.0037
No	1.000	-	-	-	1.000	-	-	-
Residential area								
Capital area	1.000	-	-	-	1.000	-	-	-
Metropolitan cities	0.998	0.948	1.051	0.9439	1.007	0.960	1.056	0.7863
Provincial areas	1.178	1.129	1.231	<0.0001	1.002	0.961	1.044	0.9254
Types of main treatment institution								
General hospital	1.019	0.970	1.070	0.4598	3.111	2.949	3.282	<0.0001
Hospital	1.102	1.038	1.170	0.0015	2.471	2.312	2.641	<0.0001
Long-term care hospital	2.108	1.985	2.238	<0.0001	7.396	6.854	7.980	<0.0001
Clinic	1.000	-	-	-	1.000	-	-	-
Sex								
Male	1.000	-	-	-	1.000	-	-	-
Female	1.000	0.961	1.040	0.9953	0.969	0.934	1.004	0.0845
Age (years)								
~69	1.000	-	-	-	1.000	-	-	-
70~74	0.887	0.828	0.951	0.0007	1.149	1.079	1.222	<0.0001
75~79	0.920	0.874	0.968	0.0014	1.049	0.997	1.103	0.068
80~	0.966	0.921	1.013	0.1546	1.021	0.969	1.075	0.4408
Types of Insurance coverage								
NHI self-employed	1.178	1.095	1.268	<0.0001	1.119	1.045	1.197	0.0012
NHI employed	0.972	0.932	1.014	0.1936	0.966	0.929	1.004	0.0764
Medical-Aid	1.000	-	-	-	1.000	-	-	-
Economic status								
Low	1.000	-	-	-	1.000	-	-	-
Mid-low	0.980	0.925	1.037	0.478	1.070	1.007	1.138	0.0296
Mid	0.980	0.923	1.041	0.5179	1.003	0.955	1.055	0.8975
Mid-high	0.984	0.925	1.047	0.6071	0.990	0.939	1.044	0.7103
High	0.969	0.914	1.028	0.299	1.043	0.991	1.097	0.1086
Period after first diagnosis								
~1 y	1.403	1.329	1.482	<0.0001	2.877	2.685	3.082	<0.0001
1~2 y	1.015	0.973	1.058	0.4995	1.259	1.204	1.316	<0.0001
2~3 y	1.001	0.967	1.036	0.9451	1.120	1.073	1.169	<0.0001
3~4 y	1.025	0.998	1.053	0.0732	1.086	1.043	1.132	<0.0001
4~5 y	1.000	-	-	-	1.000	-	-	-
Types of cancer which diagnosed								
Gastric cancer	1.000	-	-	-	1.000	-	-	-
Colorectal cancer	1.086	1.033	1.142	0.0013	1.065	1.018	1.115	0.0068
Hepatocellular carcinoma	1.184	1.107	1.268	<0.0001	1.323	1.244	1.406	<0.0001
Lung cancer	1.232	1.174	1.292	<0.0001	1.188	1.132	1.247	<0.0001
Charlson Comorbidity Index (excluded Cancer)								
~2	0.552	0.530	0.574	<0.0001	0.404	0.387	0.423	<0.0001
3~5	0.769	0.741	0.799	<0.0001	0.625	0.600	0.650	<0.0001
6~	1.000	-	-	-	1.000	-	-	-
Died in the observed year								
Dead	1.000	-	-	-	1.000	-	-	-
Alive	0.398	0.386	0.412	<0.0001	0.171	0.161	0.180	<0.0001
Year	0.989	0.980	0.998	0.0	1.078	1.066	1.091	<0.0001

## Data Availability

Data is available from NHIS upon request.

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
