# Peer review of "Do Patients Residing in Provincial Areas Transport and Spend More on Cancer Treatment in Korea?"

_ijerph, 2021, doi:10.3390/ijerph18179247_

Round 1
Reviewer 1 Report
1. Although there are 3 tables and 1 Figure in your paper from “Results” (Page 4 /10), the presentation is not so clear. For example, “…The average number of days of care, healthcare costs, and cost per day are shown in the Supplementary Table S1” the last sentence in this paragraph write “Table S1”. Where is that table?
2. It’s will be much better for a reader to easily understand all in total if the comparing results show in bar chart or pie chart.
3. For “Figure 1. The difference of patient`s residing area and major treatment area”, is there any updated version? The year 2015 is 7 years back from now. Moreover, the numbers in this Figure are not so clear.
Author Response
We have attached the response file for the reviewer.

Reviewer 2 Report
The paper entitled” Do patients residing in provincial areas transport and spend more for cancer treatment in Korea?” deals with actual and very interesting topic.
However, I have the following comments that hopefully help the authors improve their paper:
- The structure (outline) of the paper could be given at the end of the introductory chapter 1
- I suggest that the authors add a research method diagram. This will provide a snapshot of the research steps followed and will help the reader in a clearer understanding of the paper.
- I suggest to the authors a section dedicated to literature review where should analyse the existing works in the way to show the gap in the literature compared to this work. For instance, it would be better if authors can have a table comparing the closely related works on various dimensions and clearly showing the contribution of the paper.
- The authors should convince the readers of this journal, that their contribution is so important. In this context, these issues deserve a deeper discussion: What are the managerial implications from this work? How decision or policy makers could benefit from this study.
- How the results of this study can be generalized to other countries. The main contribution of this work should be compared with other similar empirical studies.
- As usual a final thorough proof-reading is recommended.
I encourage the authors to think along those questions and to develop this work further along those lines.
Author Response
Revision Note for ijerph-1317561
Title: Do patients residing in provincial areas transport and spend more for cancer treatment in Korea?
First, we greatly appreciate the comments and suggestions offered by the reviewers, which we used to improve the manuscript. Our response to each comment follows, and we have attached a revision note and also highlighted the revised sections of the manuscript. Again, thank you for the valuable and helpful comments.
Answer to Reviewer #2:
The structure (outline) of the paper could be given at the end of the introductory chapter 1
I suggest that the authors add a research method diagram. This will provide a snapshot of the research steps followed and will help the reader in a clearer understanding of the paper.
Answer: Thank you for your comments. We have added a flow diagram of study methods in Appendix and added further explanation in Methods section (page 2 line 48- page 3 line 7):
This study was conducted in two stages and hence had more than one dependent variable. The first dependent variable of this study was the location of a medical institution (local hospital vs. out-of-region hospital). This was because in the first stage, this study investigated there were differences in selecting a medical institution according to where a patient resides in. The residential area of a patient and the location of medical institution visited was defined based on the 18 administrative districts of Korea. If patients spent most of his or her medical costs in an institution located outside their residential area, these patients were categorized into the ‘out-of-region hospital’ group and vice versa. In second phase, the effect of the location of medical institution (local hospital vs. out-of-region hospital), a result of patient choice, on healthcare utilization evaluated (Supplementary Figure S1). Accordingly, the outcome variables of the study were defined differently for each stage as following:
Supplementary figure S1 Flow diagram of study methods
I suggest to the authors a section dedicated to literature review where should analyse the existing works in the way to show the gap in the literature compared to this work. For instance, it would be better if authors can have a table comparing the closely related works on various dimensions and clearly showing the contribution of the paper.
Answer: Thank you for your comments. The reviewer's comments seem to be related to the lack of comparison of prior literature in the discussion part. We added the following about the difference between our results and the previous literature according to the reviewer's opinion(page 8 line 9-39).
Many studies have investigated the relationship between patient outcomes and medical travel patterns. A systematic review which investigated the effect of travel distance (distance from the patient’s residing area to the treatment facility) on patient outcomes has show mixed results, suggesting that healthcare facilities and a patient's treatment options should be considered concurrently [16]. Patients living in rural areas eligible for Medicare tended to visit generalists in their local area, whereas they tended to travel to urban areas for specialist care [17]. Another study showed that healthcare utilization was lower in patients who needed to travel longer distances for healthcare [18], whereas others concluded that healthcare utilization and costs were higher in patients who lived further from a primary care physician [19]. Physician experience is a factor that can positively affect patient outcomes [20]. For instance, patients with ovarian cancer were found to have higher rates of mortality if they resided further away from a high-volume hospital [21]. These findings suggest that the medical travel pat-terns should not be defined simply based on absolute distance, but considered ac-counting for various factors such as patient severity and the distribution of resources.
An individual’s perception of a 'better hospital' can affect patient choice because cancer is a severe disease. In Korea, many general hospitals are skewed to the capital and metropolitan areas. Hence, as a result of patient selection, cancer patients residing in rural areas may favor care in institutions located outside their residing areas. The availability of high-end surgical technologies and hospital or physician reputation are also drivers of patient mobility [22]. In terms of hospital volume, individuals living in areas without large, prestigious hospitals nearby may prefer distant hospitals [23]. In fact, the congestion of cancer patients to the capital is a well-reported phenomenon in Korea. A study on gastric cancer patients revealed that despite gastric cancer occurring evenly in all geographical areas, most patients receive treatment in large volume institutions located in the capital area [24]. A large proportion of prostate cancer patients were also analyzed to travel for medical treatment irrespective of distance in a previous study [25]. The findings of this study add evidence on the topic of medical travel by exposing that patients residing in provincial areas have a particular tendency to travel outside their residing area.
And we added the following references:
- Kelly, C.; Hulme, C.; Farragher, T.; Clarke, G. Are differences in travel time or distance to healthcare for adults in global north countries associated with an impact on health outcomes? A systematic review. BMJ open 2016, 6, e013059.
- Chan, L.; Hart, L.G.; Goodman, D.C. Geographic access to health care for rural Medicare beneficiaries. The Journal of Rural Health 2006, 22, 140-146.
- Nemet, G.F.; Bailey, A.J. Distance and health care utilization among the rural elderly. Social Science & Medicine 2000, 50, 1197-1208.
- Billi, J.E.; Pai, C.-W.; Spahlinger, D.A. The effect of distance to primary care physician on health care utilization and disease burden. Health Care Management Review 2007, 32, 22-29.
- Choi, H.; Yang, S.-Y.; Cho, H.-S.; Kim, W.; Park, E.-C.; Han, K.-T. Mortality differences by surgical volume among patients with stomach cancer: a threshold for a favorable volume-outcome relationship. World journal of surgical oncology 2017, 15, 1-9.
- Bristow, R.E.; Chang, J.; Ziogas, A.; Gillen, D.L.; Bai, L.; Vieira, V.M. Spatial analysis of advanced-stage ovarian cancer mortality in California. American journal of obstetrics and gynecology 2015, 213, 43. e41-43. e48.
The authors should convince the readers of this journal, that their contribution is so important. In this context, these issues deserve a deeper discussion: What are the managerial implications from this work? How decision or policy makers could benefit from this study.
Answer: Thank you for your comments. We have revised the discussion section as follows (page 8 line 40- page 9 line 26).
Regional disparities in health have required attention in many countries. One factor that may act to increase the health gap between regions is the uneven distribution of resources. Imbalances in resource allocation can affect access to care, which may result in increased patient movement to hospitals outside their residing area for care [17]. The results of this study reveal that an equitable distribution of medical resources is important and provides several related implications. First, a high level of patient mobility and choice can inevitably lead to a certain amount of centralization, especially for specialized cancer services. This can be influenced by various national policies on health, and a favorable volume-outcome relationship have been demonstrated in the case of cancer [26,27]. However, the higher likelihood of patients residing in provincial areas to receive treatment at hospitals located outside their residing region requires monitoring and addressment because the findings show that healthcare costs tend to be higher in patients who travel for medical care. Such travelling may incur expenses arising from repeated diagnosis and tests, along with other costs that can arise due to various travel expenses and lost opportunity time [11]. Considering that the analysis of this study only accounted for direct healthcare costs, it can be assumed that actual total costs arising from medical travel would be significantly higher than the results shown.
Second, the time required to travel longer distances can offset potential increase in patient outcomes requires further investigation. Receiving treatment at a distant hospital has been associated with reduced accessibility and increased time-to-treatment, which can escalate risk of cancer progression [13,15,28]. Such tendencies may act as a source of regional cancer disparity, particularly in individuals with financial or physical constraints, such as the elderly and individuals with comorbidity, groups which also tend to have a higher risk of perioperative mortality [29,30]. The findings together infer the need to promote optimal configuration of cancer services that can account for patient healthcare utilization patterns, patient mobility, hospital capacity, and service quality [23]. To this end, policy makers will continuously monitor the healthcare utilization patterns of cancer patients, in addition to investigating policies that incline an efficient allocation of healthcare resources.
How the results of this study can be generalized to other countries. The main contribution of this work should be compared with other similar empirical studies.
Answer: Thank you for your comments. We have already revised the discussion section according to the reviewers' opinions, and we have already presented comparisons with previous literature.
As usual a final thorough proof-reading is recommended. I encourage the authors to think along those questions and to develop this work further along those lines.
Answer: Thank you for your comments. We have carefully revised the manuscript based on your comments. And we have proofread our manuscript.

Reviewer 3 Report
I suggest resubmission after revision.
Potentially, the text may become an nice example of secondary-data analysis. However, this is only mentioned en passant. Pointing out this aspect more extensively could improve the value of the article.
In general, the article needs a language revision both in terms of English and in terms of clarity.
Below some further comments.
1. First of all, a thorough language revision is necessary before resubmission. There are quite some odd forms, which feel to be literal translations. I suggest the authors send the article to a professional, EN-native proofreader or take advantage of the journal’s language editing service. Below I report only a few cases as examples.
- P.1 :“Cancer is a leading cause of death worldwide, with it ranking as the second leading cause of mortality before 70 years of age in over 100 countries”. It feels not it could be simplified into “Cancer is a leading cause of death worldwide and the second one for people ...".
- Introduction: “…the recent improvements in survival”. The term “survival” is odd in this context.
- Introduction “…this study objected to analyze…”. The use of “objected” is rather odd in this context. Using “aimed” may be preferable.
- Introduction “Healthcare utilization was measured based total days of care…”. Something is missing. Probably “…measured based ON THE total…”
- Section 2.1.: “…which collected based on Korean National Health Insurance (NHI) claims data.” Revise language, the phrase is not clear. Maybe a verb is missing, like “which WERE collected”.
- Section 2.2: “If patients most consumed medical cost at medical insti-tution in other area to care during year, they defined as ‘Treatment at hospital outside of residing area’. This sentence does not make sense in English.
- Section 2.2: “Considering elderly cohort, patients could be died during follow-up.” This sentence seems to be wrong in English.
- Section 2.2. “Thus, we re-calculated the medical utilization based on follow-up period by following formula.” This sentence seems wrong. It should read like “Thus, we re-calculated the medical utilization based on follow-up period BY APPLYING THE FORMULA REPORTED BELOW.
3. Some editing is also required. A few examples:
- Abstract “…their residing area Compared to….”. A full stop should go after “area”.
- Abstract “…1.093).Conclusion:…” A space should go after the full stop.
- Introduction: “…The age standardized incidence and mortality rate was…”. The verb “was “ should be changed into “were” (plural).
4. The article seems to adopt a rather odd terminology for statistical analysis. The term “interesting variables” most likely refers to the so-called "variables of interest".
5. Section 2.4. “Economic status was calculated by insurance premium which was paid by individual`s economic level, and classified as follows: ~30 (low), 31 to 60 (mid-low), 61 to 80 (mid), 81 to 90 (mid-high), and 91+ percentiles (high).”
What is this classification based upon? Does it follow NIH data or some other reference? If so, please explain. Alternatively, if based on an arbitrary classification by the authors, it should be discussed in detail.
6. Page 4: “…44,530 (69.0%) participants vice versa”. Please explain "vice versa". It is not clear.
7. The discussion of the data (section 4) is rather scant, and it is mostly a collection of references to other works and results. While it is good to show how the results of this analysis link to results of other researchers, it is necessary to have a clearer, more extensive and more focused discussion of the results of this research.
8. The Conclusion section should be elaborated further. Moreover, some comments the authors make in the Discussion section would be more suitable in the conclusions.
Author Response
Revision Note for ijerph-1317561
Title: Do patients residing in provincial areas transport and spend more for cancer treatment in Korea?
First, we greatly appreciate the comments and suggestions offered by the reviewers, which we used to improve the manuscript. Our response to each comment follows, and we have attached a revision note and also highlighted the revised sections of the manuscript. Again, thank you for the valuable and helpful comments.
Answer to Reviewer #3:
- First of all, a thorough language revision is necessary before resubmission. There are quite some odd forms, which feel to be literal translations. I suggest the authors send the article to a professional, EN-native proofreader or take advantage of the journal’s language editing service. Below I report only a few cases as examples.
P.1 :“Cancer is a leading cause of death worldwide, with it ranking as the second leading cause of mortality before 70 years of age in over 100 countries”. It feels not it could be simplified into “Cancer is a leading cause of death worldwide and the second one for people ...".
⇒ Cancer is a leading cause of death worldwide and in many countries, the second cause of mortality in people aged below 70 years of age (page 1 line 2-3)
Introduction: “…the recent improvements in survival”. The term “survival” is odd in this context.
⇒ survival rate (page 1 line 10)
Introduction “…this study objected to analyze…”. The use of “objected” is rather odd in this context. Using “aimed” may be preferable.
⇒ Additionally, the aim of this this study is to analyze whether medical travel was related to levels of healthcare utilization and costs (page 2 line 21).
Introduction “Healthcare utilization was measured based total days of care…”. Something is missing. Probably “…measured based ON THE total…”
⇒ based on the (page 2 line 23)
Section 2.1.: “…which collected based on Korean National Health Insurance (NHI) claims data.” Revise language, the phrase is not clear. Maybe a verb is missing, like “which WERE collected”.
⇒ which were collected (page 2 line 30)
Section 2.2: “If patients most consumed medical cost at medical insti-tution in other area to care during year, they defined as ‘Treatment at hospital outside of residing area’. This sentence does not make sense in English.
⇒ If patients spent most of his or her medical costs in an institution located outside their residential area, these patients were categorized into the ‘out-of-region hospital’ group and vice versa. (page 3 line 2-4)
Section 2.2: “Considering elderly cohort, patients could be died during follow-up.” This sentence seems to be wrong in English.
Section 2.2. “Thus, we re-calculated the medical utilization based on follow-up period by following formula.” This sentence seems wrong. It should read like “Thus, we re-calculated the medical utilization based on follow-up period BY APPLYING THE FORMULA REPORTED BELOW.
⇒ As the elderly cohort data used in this study included individuals aged 60 years or above, we re-calculated the medical utilization based on the follow-up period by applying the formula below. (page 3 line 8-12)
Answer: Thank you for your comments. We have revised the text according to the reviewer's comments.
- Some editing is also required. A few examples:
Abstract “…their residing area Compared to….”. A full stop should go after “area”.
⇒ Of the 64,505 participants included in this study, 19,975 (31.0%) visited medical institutions located outside their residing area. Compared to individuals residing in the capital area, those living in provincial regions (OR 2.202, 95% CI 2.068-2.344) were more likely to visit medical institutions outside their residing area.
Abstract “…1.093).Conclusion:…” A space should go after the full stop.
Introduction: “…The age standardized incidence and mortality rate was…”. The verb “was “ should be changed into “were” (plural).
⇒ were
Answer: Thank you for your comments. We have revised the text according to the reviewer's comments.
- The article seems to adopt a rather odd terminology for statistical analysis. The term “interesting variables” most likely refers to the so-called "variables of interest".
Answer: Thank you for your comments. Variable of Interests are also used as interesting variables. Our manuscript has been proofread by native speakers, and we have received comments that there are no problems with expression.
- Section 2.4. “Economic status was calculated by insurance premium which was paid by individual`s economic level, and classified as follows: ~30 (low), 31 to 60 (mid-low), 61 to 80 (mid), 81 to 90 (mid-high), and 91+ percentiles (high).” What is this classification based upon? Does it follow NIH data or some other reference? If so, please explain. Alternatively, if based on an arbitrary classification by the authors, it should be discussed in detail.
Answer: Thank you for your comments. In the data we used, insurance premiums related to economic level are provided in deciles rather than continuous variables. Since the insurance premiums provided are not continuous variable, in this study, the economic level was classified into a total of five levels considering the frequency. We have revised the sentence as follows (page 3 line 38-41).
Premiums are paid according to an individual's economic status, classified based on deciles. After considering the distribution pattern of the study participants, this study classified this variable into the following groups: ~30 (low), 31 to 60 (mid-low), 61 to 80 (mid), 81 to 90 (mid-high), and 91+ percentiles (high).
- Page 4: “…44,530 (69.0%) participants vice versa”. Please explain "vice versa". It is not clear.
Answer: Thank you for your comments. ‘Vice versa’ means a visit to a medical institution located within the residence. We have revised the text as follows (page 4 line 14-18).
Of a total of 64,505 participants treated for gastric, colorectal, hepatocellular, and lung cancer, 19,975 (31.0%) visited medical institutions situated at a location different from their residing areas, whereas 44,530 (69.0%) participants visited medical institutions within their residing area.
- The discussion of the data (section 4) is rather scant, and it is mostly a collection of references to other works and results. While it is good to show how the results of this analysis link to results of other researchers, it is necessary to have a clearer, more extensive and more focused discussion of the results of this research.
Answer: Thank you for your comments. We have carefully revised the manuscript based on your comments (page 8 line 9-39).
Many studies have investigated the relationship between patient outcomes and medical travel patterns. A systematic review which investigated the effect of travel distance (distance from the patient’s residing area to the treatment facility) on patient outcomes has show mixed results, suggesting that healthcare facilities and a patient's treatment options should be considered concurrently [16]. Patients living in rural areas eligible for Medicare tended to visit generalists in their local area, whereas they tended to travel to urban areas for specialist care [17]. Another study showed that healthcare utilization was lower in patients who needed to travel longer distances for healthcare [18], whereas others concluded that healthcare utilization and costs were higher in patients who lived further from a primary care physician [19]. Physician experience is a factor that can positively affect patient outcomes [20]. For instance, patients with ovarian cancer were found to have higher rates of mortality if they resided further away from a high-volume hospital [21]. These findings suggest that the medical travel pat-terns should not be defined simply based on absolute distance, but considered ac-counting for various factors such as patient severity and the distribution of resources.
An individual’s perception of a 'better hospital' can affect patient choice because cancer is a severe disease. In Korea, many general hospitals are skewed to the capital and metropolitan areas. Hence, as a result of patient selection, cancer patients residing in rural areas may favor care in institutions located outside their residing areas. The availability of high-end surgical technologies and hospital or physician reputation are also drivers of patient mobility [22]. In terms of hospital volume, individuals living in areas without large, prestigious hospitals nearby may prefer distant hospitals [23]. In fact, the congestion of cancer patients to the capital is a well-reported phenomenon in Korea. A study on gastric cancer patients revealed that despite gastric cancer occurring evenly in all geographical areas, most patients receive treatment in large volume institutions located in the capital area [24]. A large proportion of prostate cancer patients were also analyzed to travel for medical treatment irrespective of distance in a previous study [25]. The findings of this study add evidence on the topic of medical travel by exposing that patients residing in provincial areas have a particular tendency to travel outside their residing area.
And we added the following references:
- Kelly, C.; Hulme, C.; Farragher, T.; Clarke, G. Are differences in travel time or distance to healthcare for adults in global north countries associated with an impact on health outcomes? A systematic review. BMJ open 2016, 6, e013059.
- Chan, L.; Hart, L.G.; Goodman, D.C. Geographic access to health care for rural Medicare beneficiaries. The Journal of Rural Health 2006, 22, 140-146.
- Nemet, G.F.; Bailey, A.J. Distance and health care utilization among the rural elderly. Social Science & Medicine 2000, 50, 1197-1208.
- Billi, J.E.; Pai, C.-W.; Spahlinger, D.A. The effect of distance to primary care physician on health care utilization and disease burden. Health Care Management Review 2007, 32, 22-29.
- Choi, H.; Yang, S.-Y.; Cho, H.-S.; Kim, W.; Park, E.-C.; Han, K.-T. Mortality differences by surgical volume among patients with stomach cancer: a threshold for a favorable volume-outcome relationship. World journal of surgical oncology 2017, 15, 1-9.
- Bristow, R.E.; Chang, J.; Ziogas, A.; Gillen, D.L.; Bai, L.; Vieira, V.M. Spatial analysis of advanced-stage ovarian cancer mortality in California. American journal of obstetrics and gynecology 2015, 213, 43. e41-43. e48.
.
- The Conclusion section should be elaborated further. Moreover, some comments the authors make in the Discussion section would be more suitable in the conclusions.
Answer: Thank you for your comments. We have revised the conclusion section as follows (page 9 line 38-34):
In this study, we found that patients with gastric, colorectal, hepatocellular, and lung cancer living in provincial areas most often visited medial institutions located outside their residing region for cancer treatment. Medical costs were also higher in patients receiving care at hospitals located outside their residing areas. The findings infer the importance of appropriately distributing healthcare resources as individuals living in provincial areas may experience higher barriers in accessing cancer treatment. Continued efforts should be made to reduce the regional disparities in cancer.
Round 2
Reviewer 2 Report
The manuscript has significantly improved as compared to the previous version. Indeed, the authors tried to improve it, and the main weaknesses are solved.
Thus, in my opinion, the manuscript is recommendable for publication.
Reviewer 3 Report
I thank the authors for having implemented all the changes, whih sound convinging and appropriate.
The only point left is the use of "interesting variables" for "variables of interest". I agree with the authors that the use of the former can be found in literature. However, the standard term is the latter. This is not a problem for me, as long as the journal is OK with it.